# Different TERT Expression between Colorectal Adenoma and Serrated Polyp

**DOI:** 10.3390/medicina56090463

**Published:** 2020-09-11

**Authors:** Soo-Jung Jung, Jae-Hee Park, Ilseon Hwang, Jae-Ho Lee

**Affiliations:** 1Department of Anatomy, School of Medicine, Keimyung University, Daegu 42061, Korea; soojung@dsmc.or.kr (S.-J.J.); cpr8282@dsmc.or.kr (J.-H.P.); 2Department of Pathology, School of Medicine, Keimyung University, Dongsan Medical Center, Daegu 42061, Korea; ilseon@dsmc.or.kr

**Keywords:** TERT, colorectal cancer, tubular adenoma, serrated polyp

## Abstract

*Background and Objectives:* Telomere regulation have an association with colorectal cancer. Previous studies demonstrated its implication in colorectal carcinogenesis. This study aimed to identify the role of telomerase reverse transcriptase (TERT) in colorectal carcinogenesis and determine TERT expression and their associated genes in precancerous lesions. *Materials and Methods:* TERT expression in 93 colorectal precursor lesions was analyzed. This included 61 tubular adenomas (TAs) and 32 serrated polyps (SPs). Furthermore, KRAS and BRAF gene mutations and microsatellite instability were analyzed. Statistical tests were performed to analyze the relationship between variables. *Results:* TERT expression in TAs, when compared with those observed in paired adjacent nontumor tissues, was 0.92 ± 0.78. TERT expression levels were significantly lower in SPs (0.38 ± 0.14, *p* < 0.001). KRAS and BRAF mutations were mutually exclusive in TAs and SPs (*p* < 0.001). TERT expression tended to be associated with KRAS mutations (46.7% vs. 22.0%, *p* = 0.098) and low-grade tumors (35.0% vs. 16.0%, *p* = 0.096), but this difference was insignificant. *Conclusions:* TERT expression has a pivotal role in progression to TAs in colorectal tissue. Considering the association between TERT expression and KRAS mutation, therapeutic drugs targeting this pathway can be developed for cancer prevention.

## 1. Introduction

The majority of colorectal cancers (CRC) are developed from the adenoma–carcinoma series caused by the sequential accumulation of genetic alterations [1]. Tubular adenomas (TAs) are associated with APC, KRAS, and p53 mutations, while serrated polyps (SPs) progress through microsatellite instability (MSI) and BRAF mutations [2,3,4]. Although the detail mechanisms of these progressions are unclear, precancerous lesions may harbor various genetic characteristics and carcinogenic processes.

Telomeres are nucleoprotein complexes, composed of six-bp TTAGGG repeat sequences and capped each end of the eukaryotic chromosome [5]. In normal human somatic cells, an average length of telomeres is 5–15 kilobases and they are shortened by approximately 30–200 base pairs during every cell division. By constant shortening, telomere length becomes critically short inducing replicative senescence and apoptosis [6]. Telomerase activation allows cancer cells to overcome their fate of senescence [6,7]. Telomerase is a DNA polymerase that consists of two different subunits: the telomerase reverse transcriptase (TERT) and the telomerase RNA component (TERC) [8]. TERT overexpression was found in most cancers and showed independent prognostic values in colorectal carcinoma [8,9]. To understand the role of telomeres in colorectal carcinogenesis, we determine the TERT expression and other genetic status in precancerous lesions.

In the present study, TERT mRNA expression in 93 colorectal precursor lesions, i.e., 61 TAs and 32 SPs, was analyzed. To better understand colorectal carcinogenesis, key molecular markers in CRCs, including MSIs, KRAS, and BRAF mutations, were also analyzed in these lesions.

## 2. Materials and Methods

### 2.1. Patients and Extraction

The medical records of colonoscopic polypectomies performed at the Dongsan Medical Center during 1999–2003 were reviewed to obtain precancerous lesions. Among the cases, 61 TAs and 32 SPs were included for this study. The SPs were classified between serrated adenomas and hyperplastic polyps. Among SPs, the mixed forms and traditional serrated adenomas were excluded. Exclusion criteria included previous history of surgical resection for CRCs and evidence of hereditary non-polyposis colorectal cancer (Amsterdam criteria) or familial adenomatous polyposis. Tumor area and matched non-tumor area were selected from the slides based on hematoxylin and eosin–stained sections. Then, these paraffin-embedded tissues were used for an RNA extraction kit (Qiagen, Hilden, Germany). The quantity and quality of the isolated RNA was analyzed by the NanoDrop ND-1000 spectrophotometer (Thermo Fisher Scientific, Waltham, MA, USA).

### 2.2. TERT Expression

Reverse-transcription reactions were carried out by the High-Capacity cDNA Reverse Transcription Kit (Applied Biosystems, Foster City, CA, USA). The TERT mRNA expression level was analyzed by quantitative reverse-transcription polymerase chain reaction (qRT-PCR) and Power SYBR Green master mix (Toyobo, Japan) was used. β-Globin was used as the internal control to normalize mRNA expression levels. Relative TERT expression was determined as previously described [10].

### 2.3. KRAS and BRAF Mutations

KRAS and BRAF mutations were analyzed by pyrosequencing (PyroMark Q24, Uppsala, Sweden). The primers and pyrosequencing were designed for KRAS codon 12 and 13, BRAF V600E as described previously [11,12]. The pyrosequencing reaction was performed on the PyroMark Q24 instrument using Pyro Gold Q24 reagents (Qiagen, Venlo, The Netherlands). The pyrosequencing primers were used at a final concentration of 0.3 µmol/L. The resulting data were analyzed and quantified using PyroMark Q24 software, version 2.0.6 (Qiagen, Venlo, The Netherlands).

### 2.4. Microsatellite Instability (MSI)

Though Bethesda panel was recommended for MSI analysis, BAT25 and BAT26 analysis can precisely confirm MSI status without other markers [11,12]. Therefore, we analyzed both BAT25 and BAT 26 for MSI. PCR and further analyses were carried out as previously described [11].

### 2.5. Statistical Analysis

The SPSS statistical package, version 24.0 for Windows, was used for all statistical analyses. Chi-square, Fischer’s exact tests, and Mann–Whitney T-test analyses were performed. A two-tailed *p* value of less than 0.05 was considered to indicate statistical significance.

## 3. Results

TERT expression was successfully examined by qRT-PCR in 61 TAs and 32 SPs. TERT expression was calculated in paired normal and tumor tissues. The average TERT expression levels in TAs and SPs were 0.92 ± 0.78 and 0.38 ± 0.14, respectively, compared to that of normal tissues. These results were significantly different (*p* < 0.001, Figure 1). To determine clinical characteristics of TERT expression, patients were divided into two subgroups by the median level of TERT expression. Clinicopathological characteristics of TERT expression levels are shown in Table 1. KRAS and BRAF mutations were mutually exclusive in TAs and SPs (*p* < 0.001). In TAs, TERT expression was correlated with KRAS mutation (46.7% vs. 22.0%, *p* = 0.098) and low-grade tumors (35.0% vs. 16.0%, *p* = 0.096). However, these were no significant differences. Other variables were not associated with TERT expression.

## 4. Discussion

For the first time, our study showed the clinicopathological significance of TZAP mRNA expression in CRC precancerous lesions. Previous studies showed that telomere length had an association with clinicopathological characteristics or survival rates in CRC [7,8,9]. CRC is considered a heterogeneous disease because TAs and SPs harbor different pathological and molecular features. To clarify detail mechanism of colorectal pathogenesis, we used multitier genetic approaches with TERT expression in both TAs and SPs.

Our previous study reported similar telomere lengths in TAs and SPs [13]. However, telomere shortening was associated with other genetic changes, such as KRAS and BRAF mutations or PIK3CA amplification. Extending this finding, we examined TERT expression in precancerous lesions. Interestingly, our results showed that TERT expression level was higher in TAs than SPs. However, it did not have any association with other genetic changes participating in CRC pathogenesis. Most cancers, in contrast to most normal cells, show TERT expression upregulation changing transcriptional regulation, alternate RNA splicing, and post-translational modifications such as protein phosphorylation [5,6]. Although there was no significant difference, higher TERT expression in TAs and its association with KRAS mutation may be important clues for the transition from TA to CRC. Previous study also demonstrated that KRAS mutations increased TERT expression inducing telomere elongation [14]. It is well-known that KRAS mutation is frequent in TAs, but rare in SPs [2]. Therefore, these studies support that the combination of TERT expression and KRAS mutation contribute to TA progression. Further studies with larger sample sizes are needed to clarify this hypothesis.

Recent data showed that TERT promoter mutations are commonly found in many cancers including melanomas, bladder cancers, and hepatocellular carcinomas [15,16]. These mutations represent one possible explanation for TERT expression and telomerase reactivation in cancer cells, which can ultimately lead to cell immortalization. Therefore, we also investigated TERT promoter mutations in TAs and STs; however, mutations were not detected in both precancerous lesions (data not shown). The presence of this mutation in CRC has not been reported in previous literature. However, many studies demonstrated that the telomere length is an important factor for CRC prognosis [17,18,19]. Furthermore, our previous study reported a positive correlation between TERT and telomeric zinc-finger associated protein (TZAP) expression in colon and rectal cancers using The Cancer Genome Atlas (TCGA) database [20]. TZAP also has a great role for telomere regulation, however, its expression has not been studied in colorectal precancerous lesions and cancers. Therefore, TERT expression by other genes or environments may influence colorectal carcinogenesis. Although telomere regulation in CRC is still controversial, it should be considered further with multifactorial gene analysis.

## 5. Conclusions

In conclusion, we confirmed that TERT expression has a possible role in the tubular adenoma-carcinoma pathway. Moreover, an association between TERT expression and KRAS mutation was shown in TAs, suggesting the development of drugs targeting this pathway for cancer prevention. Therefore, further studies involving more clinical cases are needed to clarify these molecular mechanisms.

## Figures and Tables

**Figure 1 medicina-56-00463-f001:**
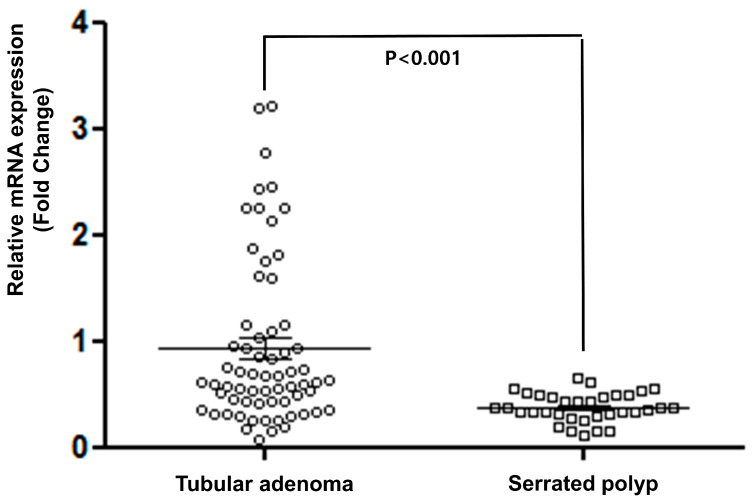
Different telomerase reverse transcriptase (TERT) expression in tubular adenoma and serrated polyp.

**Table 1 medicina-56-00463-t001:** Characteristics of TERT expression in tubular adenoma and serrated polyp.

	Tubular Adenoma (N, %)			Serrated Polyp		
	TERT (+)	TERT (−)	*p*	TERT (+)	TERT (−)	*p*
Total	18 (27.7)	47 (72.3)		6 (18.8)	26 (81.2)	
Age			0.785			0.625
<65	10 (26.3)	28 (73.7)		4 (16.7)	20 (83.3)	
≥65	8 (29.6)	19 (70.4)		2 (25.0)	6 (75.0)	
Sex			0.241			0.637
Male	10 (22.7)	34 (77.3)		5 (22.7)	17 (77.3)	
Female	8 (38.1)	13 (61.9)		1 (10.0)	9 (90.0)	
Region			0.992			0.590
Right	5 (27.8)	13 (72.2)		2 (28.6)	5 (71.4)	
Left	13 (27.7)	34 (72.3)		4 (16.0)	21 (84.0)	
KRAS mutation			0.098			1.00
(+)	7 (46.7)	8 (53.3)		0 (0)	1 (100)	
(−)	11 (22.0)	39 (78.0)		6 (19.4)	25 (80.6)	
BRAF mutation						1.00
(+)	0 (0)	0 (0)		1 (16.7)	5 (83.3)	
(−)	18 (27.7)	47 (72.3)		5 (19.2)	25 (80.8)	
MSI			0.663			0.566
(+)	1 (14.3)	6 (85.7)		0 (0)	4 (100.0)	
(−)	17 (29.3)	41 (70.7)		6 (21.4)	22 (78.6)	

MSI: microsatellite instability, TERT: telomerase reverse transcriptase.

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
