# Peer review of "Different TERT Expression between Colorectal Adenoma and Serrated Polyp"

_medicina, 2020, doi:10.3390/medicina56090463_

Round 1

Reviewer 1 Report

This is a well-designed article concerning on different TERT Expression between Colorectal 2 Adenoma and Serrated Polyp
especially that existing literature data is limited and contradictory. Authors’ findings may indicate new insights into the patomechanism of colorectal cancer and therefore may indicate a potential novel target for the treatment of colorectal cancer, which is one of the most commonly diagnosed malignancies of the digestive tract.
However, authors have to address following minor points:

Line 21 – Tas – should be TAs

Line 34,35 – tubular adenomas (TA) should be TAs, similarly in the case of serrated polyps (SP) – should be SPs

Line 46 – Which types of cancers? I suggest to mention about it.

Line 88 – According to my knowledge the Mann-Whitney U test is non-parametric test used to compare two groups but not to analyse the relationships between variables.

Line 106 – Figure description - Different TERT expression in tubular adenoma and serrated polyp –should be tubular adenomas and serrated polyps. The figure description is not consistent with table description.

In discussion I suggest to add more information about significance of genetic alterations in colon carcinogenesis. Moreover, discussion is too short.

Author Response

First of all we would like to thank the referees and editor for their time in reviewing our manuscript. Their constructive comments helped us to improve our work. We submit the revised version for your consideration. Please reconsider below a detailed response to your comments. We hope that our revised manuscript will be accepted for publication.  

Review Report (Reviewer 1)

This is a well-designed article concerning on different TERT Expression between Colorectal 2 Adenoma and Serrated Polyp
especially that existing literature data is limited and contradictory. Authors’ findings may indicate new insights into the patomechanism of colorectal cancer and therefore may indicate a potential novel target for the treatment of colorectal cancer, which is one of the most commonly diagnosed malignancies of the digestive tract.
However, authors have to address following minor points:

Line 21 – Tas – should be TAs

--> It was revised.

Line 34,35 – tubular adenomas (TA) should be TAs, similarly in the case of serrated polyps (SP) – should be SPs

--> It was revised.

Line 46 – Which types of cancers? I suggest to mention about it.

--> It was added as “in colorectal carcinoma”.

Line 88 – According to my knowledge the Mann-Whitney U test is non-parametric test used to compare two groups but not to analyse the relationships between variables.

--> It was revised.

Line 106 – Figure description - Different TERT expression in tubular adenoma and serrated polyp –should be tubular adenomas and serrated polyps. The figure description is not consistent with table description.

--> It was revised as consistently.

In discussion I suggest to add more information about significance of genetic alterations in colon carcinogenesis. Moreover, discussion is too short.

--> We added recent studies and results. Then, their significance were discussed.

Reviewer 2 Report

This paper, although of limited scope, can be of interest to researchers and clinicians dealing with colorectal cancer. There are some revisions required:

  1. Please describe how the RNA was isolated.  Writing "...the selected areas from paraffin-embedded tissues were used for RNA extraction according to the manufacturer’s instructions" is not helpful when the reader doesn't know what method was used and who the "manufacturer" is.

2. Was the RNA treated with DNase?  What was done to eliminate potential genomic DNA contamination?  Was there a negative control done for the RT-PCR to ensure lack of genomic DNA contamination, such as a no-RT control?

3. To what extent is beta-globin expressed in normal colon tissue?  Is expression constant between normal and neoplastic tissue?

4. Maybe I am mussing something, but it is not clear to me how TERT is important in the TA CRC pathway just because expression is higher in TA than SP, if the expression of TA vs. normal tissue is not significantly different.   You write: "Although there was no significant difference, higher TERT 155 expression in TA and its association with KRAS mutation may be important clues for the transition 156 from TA to CRC."  That necessitates more explanation. Is it that the maintenance of TERT along with the other mutations in TA is more oncogenic tham lower TERT and mutations in SP?  If so, it would seem the important thing to clarify is why TERT expression is lower in SP?  If my understanding is incorrect, then perhaps the Discussion needs to be more clear on these issues. It is difficult to discern what the major point is here.

Author Response

First of all we would like to thank the referees and editor for their time in reviewing our manuscript. Their constructive comments helped us to improve our work. We submit the revised version for your consideration. Please reconsider below a detailed response to your comments. We hope that our revised manuscript will be accepted for publication.  

Review Report (Reviewer 2)

This paper, although of limited scope, can be of interest to researchers and clinicians dealing with colorectal cancer. There are some revisions required:

1. Please describe how the RNA was isolated.  Writing "...the selected areas from paraffin-embedded tissues were used for RNA extraction according to the manufacturer’s instructions" is not helpful when the reader doesn't know what method was used and who the "manufacturer" is.

--> “according to the manufacturer’s instructions” was deleted because it is not helpful.

2. Was the RNA treated with DNase?  What was done to eliminate potential genomic DNA contamination?  Was there a negative control done for the RT-PCR to ensure lack of genomic DNA contamination, such as a no-RT control?

--> DNase treatment is unnecessary for this kit. However, this confirm process for contamination was performed in preliminary tests and we confirmed no genomic DNA contamination. Most recent study did not contain this preliminary study process.  However, we will add this description if necessary 

3. To what extent is beta-globin expressed in normal colon tissue?  Is expression constant between normal and neoplastic tissue?

--> beta-globin expression in normal colon tissue was also examined and its level was not significantly different between normal and neoplastic tissues.

4. Maybe I am mussing something, but it is not clear to me how TERT is important in the TA CRC pathway just because expression is higher in TA than SP, if the expression of TA vs. normal tissue is not significantly different.   You write: "Although there was no significant difference, higher TERT 155 expression in TA and its association with KRAS mutation may be important clues for the transition 156 from TA to CRC."  That necessitates more explanation. Is it that the maintenance of TERT along with the other mutations in TA is more oncogenic tham lower TERT and mutations in SP?  If so, it would seem the important thing to clarify is why TERT expression is lower in SP?  If my understanding is incorrect, then perhaps the Discussion needs to be more clear on these issues. It is difficult to discern what the major point is here.

--> We added previous studies supporting our hypothesis that TERT and KRAS co-work for TA progression. There is rare study about precancerous legions. Therefore, its further study should be continued.